# Prevalence and associated factors of anemia during pregnancy at a tertiary hospital in Zambia

Lukundo Siame [1,2,3,4]*, Francis Lwito Chishimba[1], Collins Mukubesa[2], Musonda Musonda[2], Fred Kabati[2], Michelo H. Miyoba[1], Kingsley Kamvuma[1], Sepiso K. Masenga[1,3,4]*, Benson M. Hamooya[5]

1 Department of Pathology and Microbiology, Mulungushi University, School of Medicine and Health Sciences, Livingstone, Zambia, 2 Department of Obstetrics and gynecology, Livingstone University Teaching Hospital, Livingstone Zambia, 3 Department of Cardiovascular Science and Metabolic diseases, Livingstone Center for Prevention and Translational Science, Livingstone, Zambia, 4 Department of Pathology, Livingstone Center for Prevention and Translational Science, Livingstone, Zambia, 5 Department of Public Health, Mulungushi University, School of Medicine and Health Sciences, Livingstone, Zambia

* lukundosiame23@gmail.com (LS); sepisomasenga@lcpts.org (SKM)

## Abstract

### Background

Anemia during pregnancy is among the leading causes of poor outcomes among mothers and neonates in low-resource settings like Zambia. This study aimed to determine the prevalence and factors associated with anemia among pregnant women attending antenatal care (ANC) at Livingstone University Teaching Hospital (LUTH).

### Methods

A retrospective cross-sectional study was conducted of which 307 records using a systemic random sampling method were abstracted among pregnant women attending ANC from 12th October 2023–12th January 2024. Demographic and clinical factors were collected from medical records using the Kobo Toolbox application and analyzed using STATA version 15. Descriptive statistics were used to summarize participant characteristics, and multi-variable logistic regression was employed to identify factors associated with anemia.

### Results

The median age of the participants was 26 years (interquartile range 21, 32) and the overall prevalence of anemia was 42.7% (mild: 16.8%, moderate: 20.8%, and severe: 5.1%). Factors positively associated with anemia were being in the second and third trimesters (adjusted odds ratio (AOR) =9.06, 95% confidence interval (CI): 1.72, 47.79, p = 0.009 and AOR = 4.14, 95% CI: 1.05, 16.31, p = 0.042, respectively), history

**Data availability statement:** All relevant data are within the paper and its Supporting information files.

**Funding:** The author(s) received no specific funding for this work.

**Competing interests:** The authors have declared that no competing interests exist.

of abnormal uterine bleeding (AOR = 4.09, 95% CI: 1.14, 14.72, p = 0.031), high parity (AOR = 2.09, 95% CI: 1.19, 1.61, p = 0.025), and having an underlying medical condition (AOR = 2.23, 95% CI: 1.13, 4.42, p = 0.021).

## Conclusion

Anemia in pregnancy is common in our setting, associated with being in an advanced gestational age, having a history of abnormal uterine bleeding, having high parity, and having underlying medical conditions. Targeted interventions in the second and third trimesters and among women with underlying medical conditions are imperative in reducing the burden of anemia in pregnancy and improving maternal and neonatal outcomes.

## Introduction

According to the World Health Organization (WHO) in 2019, approximately 36.5% of pregnant women had anemia globally, while the prevalence was highest in SSA standing at approximately 46.2% [1]. Anemia during pregnancy is defined as a hemoglobin concentration of less than 11g/dl according to the World Health Organization (WHO) [2]. The prevalence of anemia among pregnant women in Zambia is around 39.3% standing slightly above the global average [3,4]. Anemia in pregnancy is associated with poor maternal and neonatal outcomes such as preterm birth, low birth weight, postpartum hemorrhage, and increased maternal mortality [3,4]. This makes it a significant public health concern.

The determinants of anemia in pregnancy include iron deficiency, parasitic infections (malaria), hookworm, schistosomiasis, and HIV [1]. Other associated factors, such as high parity, short inter-pregnancy intervals, low intake of vegetables, low income, and low antenatal care utilization, also contribute to the development of anemia during pregnancy [5,6].

Tackling the high burden of anemia is pivotal to improving pregnancy outcomes [7]. However, in Zambia, where maternal and neonatal mortality remain high at 134.7 per 100,000 live births and 24.1 per 1000 live births respectively [5], there are limited studies quantifying the burden and identifying the factors associated with anemia in pregnancy. Therefore, this study aims to determine the prevalence of anemia, and the factors associated with it among pregnant women attending antenatal care (ANC) at the largest tertiary healthcare institution in the southern part of Zambia.

## Methods

### Study design and setting

This was a retrospective cross-sectional study conducted at the Livingstone University Teaching Hospital (LUTH) among pregnant women attending the Antenatal Clinic (ANC). Data abstraction was done between 21st March 2024–17th May 2024. The clinic is part of the Obstetrics and Gynecology department and serves as the main referral center for surrounding clinics in the district.

## Eligibility and sampling method

We abstracted hospital records from participants aged 18 and above who visited the ANC between 12th October 2023 and 12th January 2024. Records missing clinical and laboratory data were excluded. We used a systematic random sampling method to select every third file from the patient records in the department for screening and eligibility, see Fig 1. The data from chosen records were subsequently entered into the Kobo toolbox application (kobotoolbox.org).

## Sample size

The estimated minimal sample size was 294. We estimated a prevalence of 36.2% based on a study conducted in Zambia [6]. The alpha level was set at 5.5%, and the design effect of 1. We used OpenEpi software (openepi.com) to calculate the sample size.

## Study variables

The dependent variable in this study was anemia in pregnancy, defined according to WHO as hemoglobin < 11 g/dL in the first and third trimester and the second trimester < 10.5g/dl [2,8]. Mild anemia was defined as a hemoglobin level of 10.0 g/dL to 10.9, while moderate and severe anemia were 7.0 g/dL to 9.9 g/dL and less than 7.0 g/dL, respectively [9]. The independent variables were sociodemographic characteristics (age, religion, residence, occupation), obstetric characteristics (parity, gravidity, birth interval, trimester, abnormal uterine bleeding, history of abortions, history of Contraception use, history of antepartum hemorrhage, history of postpartum hemorrhage, history of caesarian section, underlying medical conditions, history of deworming, history of malaria prophylaxis and the use of hematinics). Some of the definitions used include the following: Nulliparous: A woman who has never given birth to a potentially viable baby (>24 weeks of gestation) is considered nulliparous; Low multiparity: Less than 5 births with gestation periods of at least 20 weeks; Grand multiparity: A woman who has had 5 or more births with gestation periods of at least 20 weeks; Gravidity: The number of times that a female has been pregnant; Primigravida: A woman who is pregnant for the first time; Multigravida: A woman

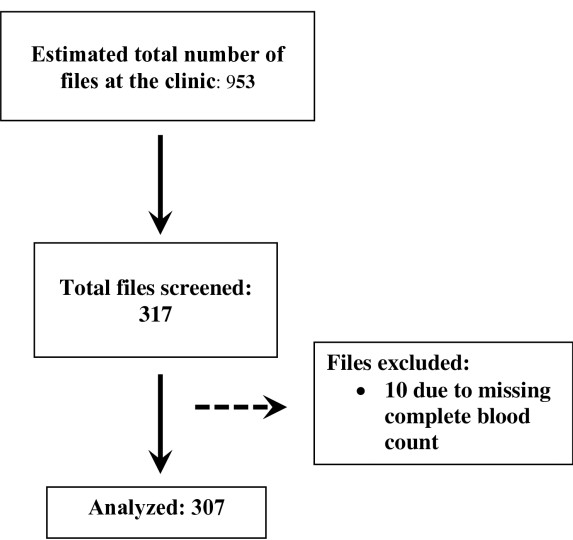

**Fig 1. Flow chart for screened and eligible files.**

---

who has been pregnant two or more times [10]. A medical condition in this study referred to a disease, disorder, or illness that is primarily treated through non-surgical means, and these included HIV, Asthma, hypertension, urinary tract infection, epilepsy, and tuberculosis.

## Data collection

Data were extracted from medical records of pregnant women attending the ANC. Trained research assistants collected the data, which were subsequently reviewed for accuracy and completeness by senior data abstractors.

## Data analysis

Data were exported to Microsoft Excel 2013 for cleaning. Statistical analyses were conducted using STATA version 15. Categorical variables were summarized with frequencies and percentages, while continuous variables were described by the median and interquartile range (IQR). The Shapiro-Wilk test was used to assess normality. The chi-square test was used to examine relationships between the outcome variable and categorical independent variables, and the Wilcoxon rank-sum test was used to compare medians. Both bivariable and multivariable logistic regression analyses were employed to identify factors associated with anemia in pregnancy. In bivariable analysis, variables with p-values less than 0.20 and those supported by the literature were included in the multivariable model to avoid overfitting. A p-value of less than 0.05 was considered statistically significant. Both crude (unadjusted) and adjusted odds ratio and their 95% CI were reported.

## Ethics

This study was approved by the Mulungushi University School of Medicine and Health Science Research Ethics Committee on June 24, 2023 (reference number SMHS-MU2-2023-159).To ensure confidentiality, all data were de-identified during data collection. As this study utilized secondary data, written or verbal consent was waived by the ethics committee. To enhance reporting, we followed the Strengthening the Reporting of Observational Studies in Epidemiology (STROBE) guidelines (S1 Strobe).

## Results

### Basic characteristics of the study participants

The median age of the participants was 26 (interquartile range (IQR): 21, 32) and the majority were protestants (n = 246, 80.4%). Most of the participants were from urban areas (n = 199, 64.8%) and were not employed (n = 211, 70.6%). Fifty-six percent (n = 172, 56.0%) of participants were low multiparous, and sixty-three-point eight percent (n = 196, 63.8%) were low multigravida. Most participants (n = 195, 63.5%) had a birth interval of more than two years and were in the third trimester (n = 268, 87.3%). Forty-two percent (n = 64, 42.7%) of the participants were diagnosed with moderate anemia and a few of them were diagnosed with abnormal uterine bleeding (n = 13, 4.3%). The majority of the participants had no history of abortions (n = 37, 87.9%). About forty-one percent (41.2%, n = 124) of participants had a history of contraceptive use. The majority of the participants had no history of antepartum (n = 296, 97.0%) or postpartum (n = 304, 99.4%) complications. Few participants (n = 25, 8.2%,) had a history of cesarean section. Twelve-point seven percent (n = 39, 12.7%) of the participants were living with HIV. The majority of participants had a history of deworming (n = 233, 76.4%) and malaria prophylaxis (n = 161, 52.6%) and nearly all participants were on hematinic (n = 298, 98.7%). The median hemoglobin of the participants was 11.3 (IQR: 9.8, 12.5) Table 1.

### Relationship between Anemia and demographic and clinical characteristics of participants

The overall prevalence of anemia was 42.7% (n = 131, 95% CI: 0.37, 0.48). The anemic participants were older than those who were non-anemic (29 years vs. 25 years, p < 0.001), see Fig 2. Anemia was more common in individuals with

**Table 1. Basic demographic and Clinical Characteristics sorted according to anemia status.**

| Variable | Median, (IQR) OR Frequency (%) | Anemia | | P value |
| --- | --- | --- | --- | --- |
| | | Yes = 131(42.7) | No = 176(57.3) | |
| Age, years, m | 26 (21, 32) | | | |
| Religion | | | | |
| Catholic | 60 (19.6) | 26(19.8) | 34(19.4) | |
| Protestant | 246 (80.4) | 105(80.2) | 141(80.6) | |
| Residence | | | | 0.344 |
| Urban | 199 (64.8) | 81(61.8) | 118(67.0) | |
| Rural | 108 (35.2) | 50(38.2) | 58(32.0 | |
| Occupation | | | | 0.182 |
| Formal | 22 (7.4) | 10(8.0) | 12(7.0) | |
| Informal | 66 (22.0) | 34(27.0) | 32(18.4) | |
| Unemployed | 211 (70.6) | 82(65.0) | 129(74.6) | |
| Parity | | | | **<0.001** |
| Null parity | 112 (36.5) | 15 (11.5) | 81(46.0) | |
| Low multiparity | 172 (56.0) | 85 (64.9) | 87(49.4) | |
| Grand multiparity | 23 (7.5) | 31 (23.6) | 8 (4.6) | |
| Gravidity | | | | **<0.001** |
| Primigravida | 111 (36.2) | 33(25.2) | 78(44.3) | |
| Multigravida | 196 (63.8) | 98(74.8) | 98(55.7) | |
| Birth interval | | | | **<0.001** |
| <two years | 112 (36.5) | 34(26.0) | 78(44.3) | |
| > two years | 195 (63.5) | 97(74.0) | 98(55.7) | |
| Trimester | | | | 0.104 |
| 1st | 19 (6.2) | 5(3.8) | 14(7.9) | |
| 2st | 20 (6.5) | 12(9.2) | 8(4.6) | |
| 3st | 268 (87.3) | 114(87.0) | 154(87.5) | |
| Abnormal uterine bleeding | | | | **0.047** |
| Yes | 13 (4.3) | 9(7.0) | 4(2.3) | |
| No | 290 (95.7) | 120(93.0) | 170(97.0) | |
| History of abortions | | | | 0.969 |
| Yes | 37 (12.1) | 115(87.8) | 153(87.9) | |
| No | 268 (87.9) | 16(12.2) | 21(12.1) | |
| History of Contraceptive | | | | **<0.001** |
| Yes | 124 (41.2) | 58(46.0) | 119(68.0) | |
| No | 177 (58.8) | 68(54.0) | 56(32.0) | |
| History of antepartum complication | | | | 0.426 |
| Yes | 9 (3.0) | 5(4.0) | 4(2.3) | |
| No | 296 (97.0) | 125(96.0) | 171(97.7) | |
| History of postpartum complication | | | | 0.829 |
| Yes | 2 (0.6) | 1(0.8) | 1(0.6) | |
| No | 304 (99.4) | 129(99.2) | 175(99.4) | |
| History of C/S | | | | **<0.001** |
| Yes | 25 (8.2) | 112(85.5) | 168(96.5) | |
| No | 280 (91.8) | 19(14.5) | 6(3.5) | |
| Medical condition | | | | **0.002** |
| HIV | 39 (12.7) | 27 (20.6) | 12 (6.8) | |

*(Continued)*

**Table 1.** (Continued)

| Variable | Median, (IQR) OR Frequency (%) | Anemia | | P value |
|---|---|---|---|---|
| | | Yes = 131(42.7) | No = 176(57.3) | |
| Other | 10 (3.3) | 100 (76.3) | 158 (89.8) | |
| None | 258 (84.0) | 4 (3.1) | 6(3.4) | |
| History of deworming | | | | 0.932 |
| Yes | 233 (76.4) | 31(23.9) | 41(23.4) | |
| No | 72 (23.6) | 99(76.1) | 134(76.6) | |
| History of malaria prophylaxis | | | | 0.17 |
| Yes | 161 (52.6) | 63(48.1) | 77(44.0) | |
| No | 145 (47.4) | 68(51.9) | 98(56.0) | |
| Hematinic | | | | 0.788 |
| Yes | 298 (98.7) | 129(98.5) | 169(98.8) | |
| No | 4(1.3) | 2(1.5) | 2(1.2) | |
| Anemia | | | | |
| Mild | 52 (39.4) | | | |
| Moderate | 64 (48.5) | | | |
| Severe | 16 (12.1) | | | |
| HB, mg/dl | 11.3 (9.8, 12.5) | | | |

Abbreviation: Hemoglobin (HB), C/S (cesarean section), other (Asthma, hypertension, urinary tract infection, epilepsy, tuberculosis).

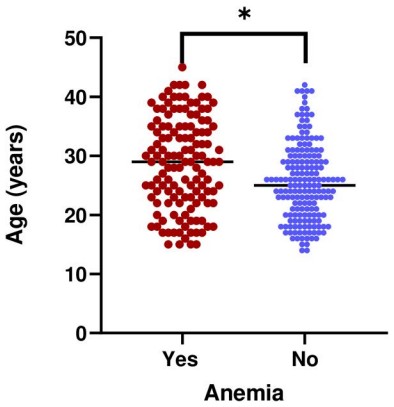

**Fig 2. Comparison of age between individuals with (n = 131) and without (n = 176) anemia.** *p < 0.001.

low multiparity compared to those with grand multiparity and nulliparity (64.9% vs. 23.7%, 11.5%). Multigravida women had a higher prevalence of anemia than primigravid women (74.8% vs. 25.2%). Women with a birth interval of more than two years were more likely to be anemic compared to those with intervals under two years (74.0% vs. 26.0%). Abnormal uterine bleeding history was more frequent in anemic individuals than those without (7.0% vs. 2.3%, p = 0.047). Anemia was more common in those participants without a history of contraceptives than in those with a history of use (54.0% vs. 46.0%). Anemia was more common with participants who had a history of Caesarean section than those without (85.5% vs. 14.5%). A significantly higher proportion of women with HIV had anemia in comparison to those without anemia, 20.6% vs. 6.8%; p = 0.002, see Table 1.

## Logistic regression analysis of factors associated with anemia

Table 2 shows factors associated with anemia in pregnancy at bivariable and multivariable logistic regression among patients. At bivariable analysis, a year increase in age was significantly associated with 7% increased odds of having anemia, odds ratio (OR) 1.07; $p < 0.001$. Those participants who were in the second trimester were 4.19 times more likely to have anemia than those in the first trimester. Participants with a history of contraceptive use had a 2.49 increased chance of having anemia compared to those without a history of contraceptive use. An increase in gravidity and parity was significantly associated with 31% times and 39% increased odds of having anemia participants, respectively. Participants with medical conditions were 2.49 times more likely to have anemia compared to those without a medical condition.

At multivariable analysis, participants in the second and third trimesters were about 9.06 times and 4.14 times, respectively, more likely to be anemic than those in the 1st trimester (adjusted odds ratio (AOR) =9.06, 95% confidence interval (CI): 1.72, 47.79, $p = 0.009$ and AOR = 4.14, 95% CI: 1.05, 16.31, $p = 0.042$, respectively), Participants with high parity were 2.09 times more likely to be anemic when compared to those with lower parity (AOR = 2.09, 95% CI 1.09, 4.42, $p = 0.025$). Women with a medical condition had 2.23 higher odds of having anemia compared to those without a medical condition (AOR = 2.23, 95% CI: 1.13, 4.42, $p = 0.021$).

## Discussion

This study aimed to determine the prevalence of anemia in pregnancy and factors associated with women attending antenatal care at Livingstone University Teaching Hospital. The prevalence of anemia was 42.7%, which is similar to a study conducted in East Africa, which found an overall prevalence of 41.82% [11]. This prevalence is particularly high when compared to other regions of the world, such as Canada (16.0%), the United States (11.5%), and China (18.5%), but it is lower than in India (50.1%) [12]. This high prevalence may stem from the low consumption of iron-rich foods in our setting, highlighting the need for targeted intervention particularly related to diet [13].

**Table 2. Bivariable and multivariable analyses of factors associated with anemia.**

| Variable | Bivariable | | Multivariable | |
| --- | --- | --- | --- | --- |
| | OR (95%) | P-value | AOR (95%, CI) | P-value |
| Age (years) | 1.07 (1.03, 1.10) | **< 0.001** | 1.01(0.95 1.06) | 0.823 |
| Trimester | | | | |
| 1st | Ref | | ref | |
| 2st | 4.19 (1.08, 16.32) | **0.038** | 9.06 (1.72, 47.79) | **0.009** |
| 3st | 2.09 (0.73, 5.96) | 0.17 | 4.14 (1.05,16.31) | **0.042** |
| Abnormal uterine bleeding | | | | |
| No | Ref | | Ref | |
| Yes | 3.19 (0.96,10.59) | 0.058 | 4.09 (1.14, 14.72) | **0.031** |
| History of contraceptive | | | | |
| No | Ref | | Ref | |
| Yes | 2.49(1.55, 3.99) | **< 0.001** | 1.44 (0.81, 2.56) | 0.216 |
| Gravity | 1.31 (1.14, 1.51) | **< 0.001** | 0.63(0.34,1.17) | 0.146 |
| Parity | 1.39 (1.19, 1.61) | **< 0.001** | 2.09(1.09, 4.02) | **0.025** |
| Medical Condition | | | | |
| No | Ref | | Ref | |
| Yes | 2.49 (1.36, 4.54) | **0.003** | 2.23 (1.13, 4.42) | **0.021** |

Abbreviation: OR (odds ratio); AOR (adjusted odds ratio), medical conditions (HIV, Asthma, hypertension, urinary tract infection, epilepsy, tuberculosis).

In this study, being in the second and third trimesters was associated with anemia. This result aligns with a study conducted in Ethiopia (2017) which found a similar association [14]. Anemia during later stages of pregnancy is linked to adverse outcomes for both mothers and newborns [15]. For the mothers, the risks include pre-eclampsia, antepartum hemorrhage, and postpartum hemorrhage, among others(puerperal sepsis, cardiac failure), with postpartum hemorrhage and pre-eclampsia being the most common causes of death in Zambia among women in both rural and urban areas [15]. Higher rates of low birth weight, small for gestational age, preterm delivery, stillbirth, and early neonatal death are linked to anemia in the second and third trimesters [16,17]. Thus, there is a need to provide targeted intervention to reduce anemia-associated maternal mortality and neonatal complications like in our setting, where a woman dies every 12 hours, a newborn dies every 30 minutes, and a stillbirth occurs every hour [18].

Participants with a history of abnormal uterine bleeding (AUB) were more likely to experience anemia during pregnancy. Research indicates that AUB is common among women of reproductive age [19,20]. AUB can result in significant blood loss, increasing the risk of iron deficiency anemia [21]. When these women become pregnant, the condition often worsen due to the increased iron demands in the second trimester from fetal development and the expansion of blood volume, which can lead to more severe forms of anemia [22,23].

In this study, an increase in parity was associated with a higher likelihood of anemia, consistent with findings from Khan et al. (2023) and Imai (2020) [24,25]. This is because each pregnancy depletes the mother's iron stores for the developing fetus and causes blood loss during delivery [26]. Therefore, if the mother's iron stores are not replenished, there is a high risk of anemia in subsequent pregnancies [27]. This presents a particular challenge in our setting due to high levels of maternal malnutrition and short inter-pregnancy interval [5,7]. Anemia can lead to poor outcomes for both the mother and the neonate [27].

Our study had limitations and strengths. Because this is a facility-based study, the results cannot be generalized. The retrospective nature of the study makes it impossible to infer other factors, such as nutritional status and causal relationships that might contribute to anemia. Despite the limitations, this study was able to determine the burden and factors associated with anemia in our setting; and it had a sufficient sample size.

## Conclusion

Anemia in pregnancy in our setting is common with a prevalence of 42.7% which is within the national range of 29 to 47.6 [12], and it was positively associated with being in the second and third, having AUB, high parity, and having a medical condition. As a result, there is a need to provide targeted interventions, such as awareness and knowledge of anemia among women of reproductive age, to prevent the negative outcome for both mother and their child.

## Supporting information

**S1 Strobe.**
(DOCX)

**S2 Dataset.**
(XLSX)

## Acknowledgments

The authors would like to thank Livingstone University Teaching Hospital management for having granted permission to conduct the study. We also thank the management of the Livingstone Center for Prevention and Translational Science for their unwavering support.

## Author contributions

**Conceptualization:** Lukundo Siame, Francis Lwito Chishimba, Benson M. Hamooya.

**Data curation:** Lukundo Siame, Francis Lwito Chishimba, Benson M. Hamooya.

**Formal analysis:** Lukundo Siame, Collins Mukubesa, Benson M. Hamooya.

**Investigation:** Lukundo Siame, Benson M. Hamooya.

**Methodology:** Lukundo Siame, Francis Lwito Chishimba, Benson M. Hamooya.

**Project administration:** Lukundo Siame.

**Resources:** Lukundo Siame, Benson M. Hamooya.

**Software:** Lukundo Siame, Benson M. Hamooya.

**Supervision:** Lukundo Siame, Benson M. Hamooya.

**Validation:** Lukundo Siame, Collins Mukubesa, Fred Kabati, Michelo H Miyoba, Sepiso K. Masenga, Benson M. Hamooya.

**Visualization:** Lukundo Siame, Francis Lwito Chishimba, Collins Mukubesa, Musonda Musonda, Fred Kabati, Michelo H Miyoba, Kingsley Kamvuma, Sepiso K. Masenga, Benson M. Hamooya.

**Writing – original draft:** Lukundo Siame, Francis Lwito Chishimba, Collins Mukubesa, Musonda Musonda, Fred Kabati, Michelo H Miyoba, Kingsley Kamvuma, Sepiso K. Masenga, Benson M. Hamooya.

**Writing – review & editing:** Lukundo Siame, Francis Lwito Chishimba, Collins Mukubesa, Musonda Musonda, Fred Kabati, Michelo H Miyoba, Kingsley Kamvuma, Sepiso K. Masenga, Benson M. Hamooya.

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
