## [Decision Letter · Decision Letter 0]

Dear Dr. Siame, 

Thank you for submitting your manuscript to PLOS ONE. After careful consideration, we feel that it has merit but does not fully meet PLOS ONE’s publication criteria as it currently stands. Therefore, we invite you to submit a revised version of the manuscript that addresses the points raised during the review process.

We look forward to receiving your revised manuscript.

Kind regards,

Ochuwa Adiketu Babah, M.Sc.PH (Epidemiology), FWACS, FMCOG

Academic Editor

PLOS ONE

Journal Requirements:

2. Please amend your list of authors on the manuscript to ensure that each author is linked to an affiliation. Authors’ affiliations should reflect the institution where the work was done (if authors moved subsequently, you can also list the new affiliation stating “current affiliation:….” as necessary).

3. Please ensure that you refer to Figure 1 in your text as, if accepted, production will need this reference to link the reader to the figure.

5. We note that there is identifying data in the Supporting Information file <S2_minimal data set.xlsx>. Due to the inclusion of these potentially identifying data, we have removed this file from your file inventory. Prior to sharing human research participant data, authors should consult with an ethics committee to ensure data are shared in accordance with participant consent and all applicable local laws.

-Location data

Please remove or anonymize all personal information (Age in years) ensure that the data shared are in accordance with participant consent, and re-upload a fully anonymized data set. Please note that spreadsheet columns with personal information must be removed and not hidden as all hidden columns will appear in the published file.

**Additional Editor Comments:**

Dear Authors,

Please revise the manuscript in line with the reviewer's comments. Also please ensure that all editorial comments are noted and that the manuscript conforms with the journal style.

Thank you.

Best regards,

Dr Ochuwa A. Babah

Reviewers' comments:

Reviewer's Responses to Questions

**Comments to the Author**

1. Is the manuscript technically sound, and do the data support the conclusions?

Reviewer #1: Yes

Reviewer #2: Yes

2. Has the statistical analysis been performed appropriately and rigorously?

Reviewer #1: Yes

Reviewer #2: Yes

3. Have the authors made all data underlying the findings in their manuscript fully available?

Reviewer #1: Yes

Reviewer #2: Yes

4. Is the manuscript presented in an intelligible fashion and written in standard English?

Reviewer #1: Yes

Reviewer #2: Yes

Reviewer #1: The manuscript addresses a very important and common problem. It is well written. my only comment is related to the demographic data. You included religion, does being protestant or catholic has different diet that could affect the prevalence of anaemia. If not please remove it. The same goes for the residence.

Also please add the word complications to the (history of antepartum and history of postpartum)

Reviewer #2: There is some grammar mistakes that the authors must correct. The mathematical equations of presented model were not provided. A figure that depicted the anemia in terms of ages were not presented.

**Do you want your identity to be public for this peer review?** For information about this choice, including consent withdrawal, please see our Privacy Policy

Reviewer #1: **Yes: ** Nourah Hasan Al Qahtani

Reviewer #2: No

---

## [Author Response · Author response to Decision Letter 1]

26 Dec 2024

24th December, 2024

To the reviewers and Editor,

Ref: RESPONSES TO REVIEWER COMMENTS

We would like to thank the reviewers for taking the time to make suggestions that have improved our manuscript. We have now made revisions to the comments in the manuscript and incorporated all suggestions. We now hope the current manuscript is acceptable for publication. Below are the point-by-point responses to all comments and suggestions

Editorial comments

Response: thank you. Files named according

2. Please amend your list of authors on the manuscript to ensure that each author is linked to an affiliation. Authors’ affiliations should reflect the institution where the work was done (if authors moved subsequently, you can also list the new affiliation stating “current affiliation:….” as necessary).

Response: thank you. all authors have been given an affiliations

3. Please ensure that you refer to Figure 1 in your text as, if accepted, production will need this reference to link the reader to the figure.

Response: thank you. Figure have been removed and uploaded as separate figures acceptable to the Journal requirements

Response: thank you. Supporting document caption has been updated at the end of the manuscript as per journal guidelines

5. We note that there is identifying data in the Supporting Information file <S2_minimal data set.xlsx>. Due to the inclusion of these potentially identifying data, we have removed this file from your file inventory. Prior to sharing human research participant data, authors should consult with an ethics committee to ensure data are shared in accordance with participant consent and all applicable local laws.

-Location data

Please remove or anonymize all personal information (Age in years) ensure that the data shared are in accordance with participant consent, and re-upload a fully anonymized data set. Please note that spreadsheet columns with personal information must be removed and not hidden as all hidden columns will appear in the published file.

Response: thank you. All identifying information have been removed from the data set

Response: thank you all references have been checked and cited accordingly in the text.

Reviewer Comments

Reviewer #1: The manuscript addresses a very important and common problem. It is well written.

Response: thank you for reviewing of our article

My only comment is related to the demographic data. You included religion, does being protestant or catholic has different diet that could affect the prevalence of anaemia. If not please remove it. The same goes for the residence.

Response: Thank you very much. We included religion because, in our setting, certain Protestant religious groups (for example, Seventh-day Adventists, who make up 7.1%) advocate for vegetarianism. Regarding residence, we aimed to explore whether differences exist, as rural-based individuals tend to have a predominantly vegetable-based diet when compared to the urban based individuals.

Also please add the word complications to the (history of antepartum and history of postpartum)

Response: thank you for the suggestion. We have added the word complication

Reviewer #2:

There is some grammar mistakes that the authors must correct.

Response: thank you for the observation. We have made effort to revise grammar.

The mathematical equations of presented model were not provided.

Response: Thank you. To arrive at the final model, we used variables with p-values less than 0.20 and those supported by the literature. These were included in the multivariable analysis, as explained in the methods section.

A figure that depicted the anemia in terms of ages were not presented.

Response: Thank you. A figure depicting anemia in terms of age has now been presented as Figure 2.

We have revised the manuscript and addressed all concerns raised by the reviewers. We want to thank you all again for the tremendous work and time that you committed in editing our work. Our manuscript is much improved, and we are very grateful.

Yours sincerely,

Dr. Lukundo Siame, Bsc., MBcHB.

Junior Residence Medical officer Livingstone University Teaching Hospital, Zambia

---

## [Decision Letter · Decision Letter 1]

Dear Dr. Siame,

Thank you for submitting your manuscript to PLOS ONE. After careful consideration, we feel that it has merit but does not fully meet PLOS ONE’s publication criteria as it currently stands. Therefore, we invite you to submit a revised version of the manuscript that addresses the points raised during the review process.

We look forward to receiving your revised manuscript.

Kind regards,

Jennifer Yourkavitch

Academic Editor

PLOS ONE

Journal Requirements:

**Additional Editor Comments:**

Thank you for this revision. I am a new editor recently assigned to this manuscript. I have just one last editing note:

In the Results section, please do not begin sentences with a number, e.g., 8.2% of women xxx etc. You can begin those sentences with the spelling of the number, e.g., Eight percent of women xxx etc. or reconstruct those sentences, e.g., Few women (8.2.%) were xxx etc.

Reviewers' comments:

Reviewer's Responses to Questions

**Comments to the Author**

Reviewer #2: All comments have been addressed

2. Is the manuscript technically sound, and do the data support the conclusions?

Reviewer #2: Yes

3. Has the statistical analysis been performed appropriately and rigorously?

Reviewer #2: Yes

4. Have the authors made all data underlying the findings in their manuscript fully available?

Reviewer #2: Yes

5. Is the manuscript presented in an intelligible fashion and written in standard English?

Reviewer #2: Yes

Reviewer #2: The answers to the all comments were presented and prepared. There are no any reminded comments to be answered.

**Do you want your identity to be public for this peer review?** For information about this choice, including consent withdrawal, please see our Privacy Policy

Reviewer #2: **Yes: ** Solaiman Afroughi

---

## [Author Response · Author response to Decision Letter 2]

17 Jun 2025

We would like to thank the reviewers for taking the time to make suggestions that have improved our manuscript. We have revised the manuscript and addressed all concerns and suggestions. We now hope the current manuscript is acceptable for publication. Below are the point-by-point responses to all comments and suggestions.

Response to the reviewer:

Academic editor’s comment:

Thank you for this revision. I am a new editor recently assigned to this manuscript. I have just one last editing note: In the Results section, please do not begin sentences with a number, e.g., 8.2% of women xxx etc. You can begin those sentences with the spelling of the number, e.g., Eight percent of women xxx etc. or reconstruct those sentences, e.g., Few women (8.2.%) were xxx etc.

Response: Thank you very much for the suggestion: we have revised the manuscript the manuscript as suggested.

Reviewer comments:

Reviewer #2: The answers to all comments were presented and prepared. There are no any reminded comments to be answered.

Response: thank you.

We have revised the manuscript and addressed all concerns raised. We want to thank you all again for the tremendous work and time that you committed to reviewing and correcting our work. Our manuscript is much improved, and we are very grateful.

---

## [Editor Report · Decision Letter 2]

Prevalence and associated factors of anemia during pregnancy at a tertiary hospital in Zambia

PONE-D-24-48610R2

Dear Dr. Siame,

We’re pleased to inform you that your manuscript has been judged scientifically suitable for publication and will be formally accepted for publication once it meets all outstanding technical requirements.

Kind regards,

Jennifer Yourkavitch

Academic Editor

PLOS ONE

---

## [Editor Report · Acceptance letter]

PONE-D-24-48610R2

PLOS ONE

Dear Dr. Siame,

I'm pleased to inform you that your manuscript has been deemed suitable for publication in PLOS ONE. Congratulations! Your manuscript is now being handed over to our production team.

Kind regards,

on behalf of

Dr. Jennifer Yourkavitch

Academic Editor

PLOS ONE